# Semi-supervised sequence tagging with bidirectional language models

## Abstract

Pre-trained word embeddings learned from unlabeled text have become a standard component of neural network architectures for NLP tasks. However, in most cases, the recurrent network that operates on word-level representations to produce context sensitive representations is trained on relatively little labeled data. In this paper, we demonstrate a general semi-supervised approach for adding pretrained context embeddings from bidirectional language models to NLP systems and apply it to sequence labeling tasks. We evaluate our model on two standard datasets for named entity recognition (NER) and chunking, and in both cases achieve state of the art results, surpassing previous systems that use other forms of transfer or joint learning with additional labeled data and task specific gazetteers.

## 1 Introduction

Due to their simplicity and efficacy, pre-trained word embedding have become ubiquitous in NLP systems. Many prior studies have shown that they capture useful semantic and syntactic information (Mikolov et al., 2013; Pennington et al., 2014) and including them in NLP systems has been shown to be enormously helpful for a variety of downstream tasks (Collobert et al., 2011).

However, in many NLP tasks it is essential to represent not just the meaning of a word, but also the word in context. For example, in the two phrases "A Central Bank spokesman" and "The Central African Republic", the word 'Central' is used as part of both an Organization and Location. Accordingly, current state of the art sequence tagging models typically include a bidirectional re-

current neural network (RNN) that encodes token sequences into a context sensitive representation before making token specific predictions (Yang et al., 2017; Ma and Hovy, 2016; Lample et al., 2016; Hashimoto et al., 2016).

Although the token representation is initialized with pre-trained embeddings, the parameters of the bidirectional RNN are typically learned only on labeled data. Previous work has explored methods for jointly learning the bidirectional RNN with supplemental labeled data from other tasks (e.g., Søgaard and Goldberg, 2016; Yang et al., 2017).

In this paper, we explore an alternate semi-supervised approach which does not require additional labeled data. We use a neural language model (LM) pre-trained on a large, unlabeled corpus to compute an encoding of the context at each position in the sequence (hereafter an *LM embedding*) and use it in the supervised sequence tagging model. Since the LM embeddings are used to compute the probability of future words in a neural LM, they are likely to encode both the semantic and syntactic roles of words in context.

Our main contribution is to show that the context sensitive representation captured in the LM embeddings is useful in the supervised sequence tagging setting. When we include the LM embeddings in our system overall performance increases from 90.87% to 91.93% $F_1$ for the CoNLL 2003 NER task, a more then 1% absolute F1 increase, and a substantial improvement over the previous state of the art. We also establish a new state of the art result (96.37% $F_1$) for the CoNLL 2000 Chunking task.

As a secondary contribution, we show that using both forward and backward LM embeddings boosts performance over a forward only LM. We also demonstrate that domain specific pre-training is not necessary by applying a LM trained in the news domain to scientific papers.

## 2 Language model augmented sequence taggers (TagLM)

### 2.1 Overview

The main components in our language-model-augmented sequence tagger (TagLM) are illustrated in Fig. 1. After pre-training word embeddings and a neural LM on large, unlabeled corpora (Step 1), we extract the word and LM embeddings for every token in a given input sequence (Step 2) and use them in the supervised sequence tagging model (Step 3).

### 2.2 Baseline sequence tagging model

Our baseline sequence tagging model is a hierarchical neural tagging model, closely following a number of recent studies (Ma and Hovy, 2016; Lample et al., 2016; Yang et al., 2017; Chiu and Nichols, 2016) (left side of Figure 2).

Given a sentence of tokens $(t_1, t_2, \ldots, t_N)$ it first forms a representation, $\mathbf{x}_k$, for each token by concatenating a character based representation $\mathbf{c}_k$ with a token embedding $\mathbf{w}_k$:

$$\mathbf{c}_k = C(t_k; \theta_c)$$
$$\mathbf{w}_k = E(t_k; \theta_w)$$
$$\mathbf{x}_k = [\mathbf{c}_k; \mathbf{w}_k] \tag{1}$$

The character representation $\mathbf{c}_k$ captures morphological information and is either a convolutional neural network (CNN) (Ma and Hovy, 2016; Chiu and Nichols, 2016) or RNN (Yang et al., 2017; Lample et al., 2016). It is parameterized by $C(\cdot, \theta_c)$ with parameters $\theta_c$. The token embeddings, $\mathbf{w}_k$, are obtained as a lookup $E(\cdot, \theta_w)$, initialized using pre-trained word embeddings, and fine tuned during training (Collobert et al., 2011).

To learn a context sensitive representation, we employ multiple layers of bidirectional RNNs. For each token position, $k$, the hidden state $\mathbf{h}_{k,i}$ of RNN layer $i$ is formed by concatenating the hidden states from the forward ($\overrightarrow{\mathbf{h}}_{k,i}$) and backward ($\overleftarrow{\mathbf{h}}_{k,i}$) RNNs. As a result, the bidirectional RNN is able to use both past and future information to make a prediction at token $k$. More formally, for the first RNN layer that operates on $\mathbf{x}_k$ to output $\mathbf{h}_{k,1}$:

$$\overrightarrow{\mathbf{h}}_{k,1} = \overrightarrow{R}_1(\mathbf{x}_k, \overrightarrow{\mathbf{h}}_{k-1,1}; \theta_{\overrightarrow{R}_1})$$
$$\overleftarrow{\mathbf{h}}_{k,1} = \overleftarrow{R}_1(\mathbf{x}_k, \overleftarrow{\mathbf{h}}_{k+1,1}; \theta_{\overleftarrow{R}_1})$$
$$\mathbf{h}_{k,1} = [\overrightarrow{\mathbf{h}}_{k,1}; \overleftarrow{\mathbf{h}}_{k,1}] \tag{2}$$

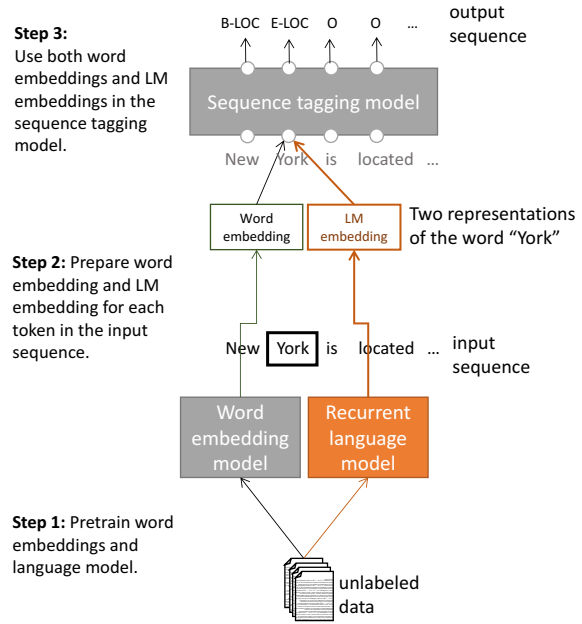

Figure 1: The main components in TagLM, our language-model-augmented sequence tagging system. The language model component (in orange) is used to augment the input token representation in a traditional sequence tagging models (in grey).

The second RNN layer is similar and uses $\mathbf{h}_{k,1}$ to output $\mathbf{h}_{k,2}$. In this paper, we use $L = 2$ layers of RNNs in all experiments and parameterize $R_i$ as either Gated Recurrent Units (GRU) (Cho et al., 2014) or Long Short-Term Memory units (LSTM) (Hochreiter and Schmidhuber, 1997) depending on the task.

Finally, the output of the final RNN layer $\mathbf{h}_{k,L}$ is used to predict a score for each possible tag using a single dense layer. Due to the dependencies between successive tags in our sequence labeling tasks (e.g. using the BIOES labeling scheme, it is not possible for `I-PER` to follow `B-LOC`), it is beneficial to model and decode each sentence jointly instead of independently predicting the label for each token. Accordingly, we add another layer with parameters for each label bigram, computing the sentence conditional random field (CRF) loss (Lafferty et al., 2001) using the forward-backward algorithm at training time, and using the Viterbi algorithm to find the most likely tag sequence at test time, similar to Collobert et al. (2011).

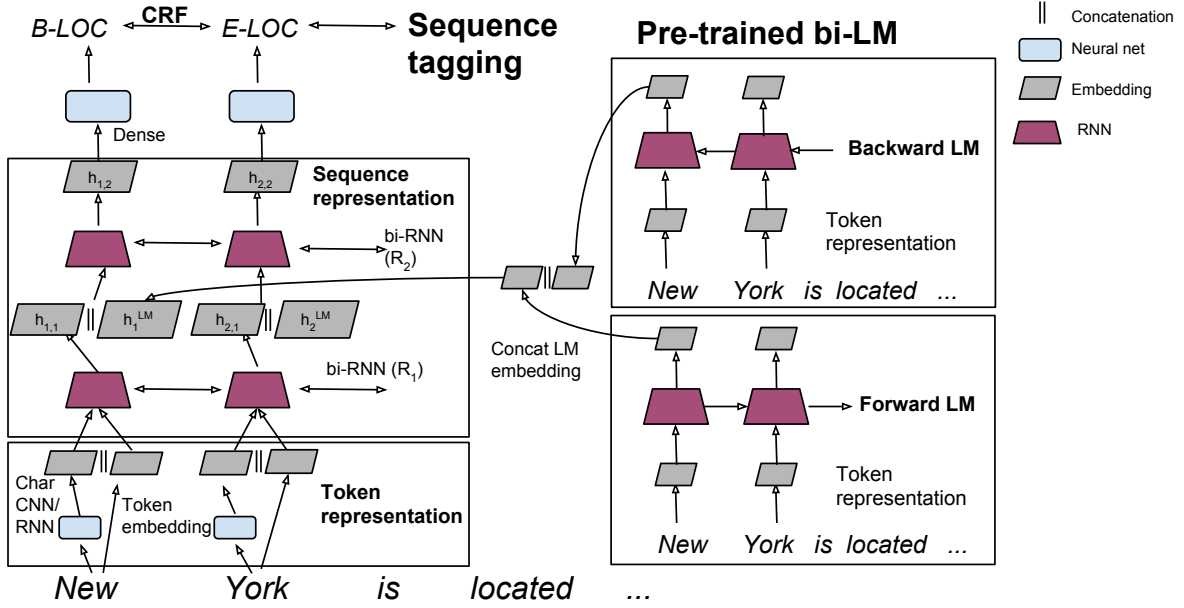

Figure 2: Overview of TagLM, our language model augmented sequence tagging architecture. The top level embeddings from a pre-trained bidirectional LM are inserted in a stacked bidirectional RNN sequence tagging model. See text for details.

## 2.3 Bidirectional LM

A language model computes the probability of a token sequence $(t_1, t_2, \ldots, t_N)$

$$p(t_1, t_2, \ldots, t_N) = \prod_{k=1}^{N} p(t_k \mid t_1, t_2, \ldots, t_{k-1}).$$

Recent state of the art neural language models (Józefowicz et al., 2016) use a similar architecture to our baseline sequence tagger where they pass a token representation (either from a CNN over characters or as token embeddings) through multiple layers of LSTMs to embed the history $(t_1, t_2, \ldots, t_k)$ into a fixed dimensional vector $\overrightarrow{\mathbf{h}}_k^{LM}$. This is the *forward LM embedding* of the token at position $k$ and is the output of the top LSTM layer in the language model. Finally, the language model predicts the probability of token $t_{k+1}$ using a softmax layer over words in the vocabulary.

The need to capture future context in the LM embeddings suggests it is beneficial to also consider a *backward* LM in additional to the traditional *forward* LM. A backward LM predicts the previous token given the future context. Given a sentence with $N$ tokens, it computes

$$p(t_1, t_2, \ldots, t_N) = \prod_{k=1}^{N} p(t_k \mid t_{k+1}, t_{k+2}, \ldots, t_N).$$

A backward LM can be implemented in an analogous way to a forward LM and produces the *backward LM embedding* $\overleftarrow{\mathbf{h}}_k^{LM}$, for the sequence $(t_k, t_{k+1}, \ldots, t_N)$, the output embeddings of the top layer LSTM.

In our final system, after pre-training the forward and backward LMs separately, we remove the top layer softmax and concatenate the forward and backward LM embeddings to form bidirectional LM embeddings, i.e., $\mathbf{h}_k^{LM} = [\overrightarrow{\mathbf{h}}_k^{LM}; \overleftarrow{\mathbf{h}}_k^{LM}]$. Note that in our formulation, the forward and backward LMs are independent, without any shared parameters.

## 2.4 Combining LM with sequence model

Our combined system, TagLM, uses the LM embeddings as additional inputs to the sequence tagging model. In particular, we concatenate the LM embeddings $\mathbf{h}^{LM}$ with the output from one of the bidirectional RNN layers in the sequence model. In our experiments, we found that introducing the LM embeddings at the output of the first layer performed the best. More formally, we simply replace (2) with

$$\mathbf{h}_{k,1} = [\overrightarrow{\mathbf{h}}_{k,1}; \overleftarrow{\mathbf{h}}_{k,1}; \mathbf{h}_k^{LM}]. \qquad (3)$$

## 3   Experiments

We evaluate our approach on two well bench-marked sequence tagging tasks, the CoNLL 2003 NER task (Sang and Meulder, 2003) and the CoNLL 2000 Chunking task (Sang and Buchholz, 2000). We report the official evaluation metric (micro-averaged $F_1$). In both cases, we use the BIOES labeling scheme for the output tags, following previous work which showed it outperforms other options (e.g., Ratinov and Roth, 2009). Following Chiu and Nichols (2016), we use the Senna word embeddings (Collobert et al., 2011) and pre-processed the text by lowercasing all tokens and replacing all digits with 0.

### 3.0.1   CoNLL 2003 NER

The CoNLL 2003 NER task consists of newswire from the Reuters RCV1 corpus tagged with four different entity types (PER, LOC, ORG, MISC). It includes standard train, development and test sets. Following previous work (Yang et al., 2017; Chiu and Nichols, 2016) we trained on both the train and development sets after tuning hyperparameters on the development set.

The hyperparameters for our baseline model are similar to Yang et al. (2017). We use two bidirectional GRUs with 80 hidden units and 25 dimensional character embeddings for the token character encoder. The sequence layer uses two bidirectional GRUs with 300 hidden units each. For regularization, we add 25% dropout to the input of each GRU, but not to the recurrent connections.

### 3.0.2   CoNLL 2000 chunking

The CoNLL 2000 chunking task uses sections 15-18 from the Wall Street Journal corpus for training and section 20 for testing. It defines 11 syntactic chunk types (e.g., NP, VP, ADJP) in addition to other. We randomly sampled 1000 sentences from the training set as a held-out development set.

The baseline sequence tagger uses 30 dimensional character embeddings and a CNN with 30 filters of width 3 characters followed by a tanh non-linearity for the token character encoder. The sequence layer uses two bidirectional LSTMs with 200 hidden units. Following Ma and Hovy (2016) we added 50% dropout to the character embeddings, the input to each LSTM layer (but not recurrent connections) and to the output of the final LSTM layer.

### 3.0.3   Pre-trained language models

The primary bidirectional LMs we used in this study were trained on the 1B Word Benchmark (Chelba et al., 2014), a publicly available benchmark for large-scale language modeling. The training split has approximately 800 million tokens, about a 4000X increase over the number training tokens in the CoNLL datasets. Józefowicz et al. (2016) explored several model architectures and released their best single model and training recipes. Following Sak et al. (2014), they used linear projection layers at the output of each LSTM layer to reduce the computation time but still maintain a large LSTM state. Their single best model took three weeks to train on 32 GPUs and achieved 30.0 test perplexity. It uses a character CNN with 4096 filters for input, followed by two stacked LSTMs, each with 8192 hidden units and a 1024 dimensional projection layer. We use CNN-BIG-LSTM to refer to this language model in our results.

In addition to CNN-BIG-LSTM from Józefowicz et al. (2016),[1] we used the same corpus to train two additional language models with fewer parameters: forward LSTM-2048-512 and backward LSTM-2048-512. Both language models use token embeddings as input to a single layer LSTM with 2048 units and a 512 dimension projection layer. We closely followed the procedure outlined in Józefowicz et al. (2016), except we used synchronous parameter updates across four GPUs instead of asynchronous updates across 32 GPUs and ended training after 10 epochs. The test set perplexities for our forward and backward LSTM-2048-512 language models are 47.7 and 47.3, respectively.[2]

### 3.0.4   Training

All experiments use the Adam optimizer (Kingma and Ba, 2014) with gradient norms clipped at 5.0. In addition to explicit dropout regularization, we also use early stopping to prevent over-fitting and use the following process to determine when to stop training. We first train with a constant learning rate $\alpha = 0.001$ on the training data and monitor the development set performance at each epoch. Then, at the epoch with the highest de-

---

[1] https://github.com/tensorflow/models/tree/master/lm_1b

[2] Due to different implementations, the perplexity of the forward LM with similar configurations in Józefowicz et al. (2016) is different (45.0 vs. 47.7).

| Model | $F_1\pm$ std |
|---|---|
| Chiu and Nichols (2016) | $90.91 \pm 0.20$ |
| Lample et al. (2016) | $90.94$ |
| Ma and Hovy (2016) | $91.37$ |
| Our baseline without LM | $90.87 \pm 0.13$ |
| TagLM | $\mathbf{91.93 \pm 0.19}$ |

Table 1: Test set $F_1$ comparison on CoNLL 2003 NER task, using only CoNLL 2003 data and unlabeled text.

| Model | $F_1\pm$ std |
|---|---|
| Yang et al. (2017) | $94.66$ |
| Hashimoto et al. (2016) | $95.02$ |
| Søgaard and Goldberg (2016) | $95.28$ |
| Our baseline without LM | $95.00 \pm 0.08$ |
| TagLM | $\mathbf{96.37 \pm 0.05}$ |

Table 2: Test set $F_1$ comparison on CoNLL 2000 Chunking task using only CoNLL 2000 data and unlabeled text.

velopment performance, we start a simple learning rate annealing schedule: decrease $\alpha$ an order of magnitude (i.e., divide by ten), train for five epochs, decrease $\alpha$ an order of magnitude again, train for five more epochs and stop.

Following Chiu and Nichols (2016), we train each final model configuration ten times with different random seeds and report the mean and standard deviation $F_1$. It is important to estimate the variance of model performance since the test data sets are relatively small.

### 3.1 Overall system results

Tables 1 and 2 compare results from TagLM with previously published state of the art results without additional labeled data or task specific gazetteers. Tables 3 and 4 compare results of TagLM to other systems that include additional labeled data or gazetteers. In both tasks, TagLM establishes a new state of the art using bidirectional LMs (the forward CNN-BIG-LSTM and the backward LSTM-2048-512).

In the CoNLL 2003 NER task, our model scores 91.93 mean $F_1$, which is a statistically significant increase over the previous best result of 91.62 $\pm0.33$ from Chiu and Nichols (2016) that used gazetteers (at 95%, two-sided Welch t-test, $p = 0.021$).

In the CoNLL 2000 Chunking task, TagLM

achieves 96.37 mean $F_1$, exceeding all previously published results without additional labeled data by more then 1% absolute $F_1$. The improvement over the previous best result of 95.77 in Hashimoto et al. (2016) that jointly trains with Penn Treebank (PTB) POS tags is statistically significant at 95% ($p < 0.001$ assuming standard deviation of 0.1).

Importantly, the LM embeddings amounts to an average absolute improvement of 1.06 and 1.37 $F_1$ in the NER and Chunking tasks, respectively.

**Adding external resources.** Although we do not use external labeled data or gazetteers, we found that TagLM outperforms previous state of the art results in both tasks when external resources (labeled data or task specific gazetteers) are available. Furthermore, Tables 3 and 4 show that, in most cases, the improvements we obtain by adding LM embeddings are larger then the improvements previously obtained by adding other forms of transfer or joint learning. For example, Yang et al. (2017) noted an improvement of only 0.06 $F_1$ in the NER task when transfer learning from both CoNLL 2000 chunks and PTB POS tags and Chiu and Nichols (2016) reported an increase of 0.71 $F_1$ when adding gazetteers to their baseline. In the Chunking task, previous work has reported from 0.28 to 0.75 improvement in $F_1$ when including supervised labels from the PTB POS tags or CoNLL 2003 entities (Yang et al., 2017; Søgaard and Goldberg, 2016; Hashimoto et al., 2016).

### 3.2 Analysis

To elucidate the characteristics of our LM augmented sequence tagger, we ran a number of additional experiments on the CoNLL 2003 NER task.

**How to use LM embeddings?** In this experiment, we concatenate the LM embeddings at different locations in the baseline sequence tagger. In particular, we used the LM embeddings $\mathbf{h}_k^{LM}$ to:

- augment the *input* of the *first* RNN layer; i.e., $\mathbf{x}_k = [\mathbf{c}_k; \mathbf{w}_k; \mathbf{h}_k^{LM}]$,

- augment the *output* of the *first* RNN layer; i.e., $\mathbf{h}_{k,1} = [\overrightarrow{\mathbf{h}}_{k,1}; \overleftarrow{\mathbf{h}}_{k,1}; \mathbf{h}_k^{LM}]$,[3] and

- augment the *output* of the *second* RNN layer; i.e., $\mathbf{h}_{k,2} = [\overrightarrow{\mathbf{h}}_{k,2}; \overleftarrow{\mathbf{h}}_{k,2}; \mathbf{h}_k^{LM}]$.

---
[3]This configuration the same as Eq. 3 in §2.4. It was reproduced here for convenience.

| Model | External resources | $F_1$ Without | $F_1$ With | $\Delta$ |
|---|---|---|---|---|
| Yang et al. (2017) | transfer from CoNLL 2000/PTB-POS | 91.2 | 91.26 | +0.06 |
| Chiu and Nichols (2016) | with gazetteers | 90.91 | 91.62 | +0.71 |
| Collobert et al. (2011) | with gazetteers | 88.67 | 89.59 | +0.92 |
| Luo et al. (2015) | joint with entity linking | 89.9 | 91.2 | **+1.3** |
| Ours | no LM vs TagLM *unlabeled data only* | 90.87 | **91.93** | +1.06 |

Table 3: Improvements in test set $F_1$ in CoNLL 2003 NER when including additional labeled data or task specific gazetteers (except the case of TagLM where we do not use additional labeled resources).

| Model | External resources | $F_1$ Without | $F_1$ With | $\Delta$ |
|---|---|---|---|---|
| Yang et al. (2017) | transfer from CoNLL 2003/PTB-POS | 94.66 | 95.41 | +0.75 |
| Hashimoto et al. (2016) | jointly trained with PTB-POS | 95.02 | 95.77 | +0.75 |
| Søgaard and Goldberg (2016) | jointly trained with PTB-POS | 95.28 | 95.56 | +0.28 |
| Ours | no LM vs TagLM *unlabeled data only* | 95.00 | **96.37** | +1.37 |

Table 4: Improvements in test set $F_1$ in CoNLL 2000 Chunking when including additional labeled data (except the case of TagLM where we do not use additional labeled data).

| Use LM embeddings at | $F_1 \pm$ **std** |
|---|---|
| input to the first RNN layer | $91.55 \pm 0.21$ |
| output of the first RNN layer | $\mathbf{91.93 \pm 0.19}$ |
| output of the second RNN layer | $91.72 \pm 0.13$ |

Table 5: Comparison of CoNLL-2003 test set $F_1$ when the LM embeddings are included at different layers in the baseline tagger.

Table 5 shows that the second alternative performs best. We speculate that the second RNN layer in the sequence tagging model is able to capture interactions between task specific context as expressed in the first RNN layer and general context as expressed in the LM embeddings in a way that improves overall system performance. These results are consistent with Søgaard and Goldberg (2016) who found that chunking performance was sensitive to the level at which additional POS supervision was added.

**Does it matter which language model to use?** In this experiment, we compare six different configurations of the forward and backward language models (including the baseline model which does not use any language models). The results are reported in Table 6.

We find that adding backward LM embeddings consistently outperforms forward-only LM embeddings, with $F_1$ improvements between 0.22 and 0.27%, even with the relatively small backward `LSTM-2048-512` LM.

LM size is important, and replacing the forward `LSTM-2048-512` with `CNN-BIG-LSTM` (test perplexities of 47.7 to 30.0 on 1B Word Benchmark) improves $F_1$ by 0.26 - 0.31%, about as much as adding backward LM. Accordingly, we hypothesize (but have not tested) that replacing the backward `LSTM-2048-512` with a backward LM analogous to the `CNN-BIG-LSTM` would further improve performance.

To highlight the importance of including language models trained on a large scale data, we also experimented with training a language model on just the CoNLL 2003 training and development data. Due to the much smaller size of this data set, we decreased the model size to 512 hidden units with a 256 dimension projection and normalized tokens in the same manner as input to the sequence tagging model (lower-cased, with all digits replaced with 0). The test set perplexities for the forward and backward models (measured on the CoNLL 2003 test data) were 106.9 and 104.2, respectively. Including embeddings from these language models *decreased* performance slightly compared to the baseline system without any LM. This result supports the hypothesis that adding language models help because they learn composi-

| Forward language model | Backward language model | LM perplexity | | $F_1 \pm$ std |
|---|---|---|---|---|
| | | Fwd | Bwd | |
| — | — | N/A | N/A | $90.87 \pm 0.13$ |
| LSTM-512-256* | LSTM-512-256* | 106.9 | 104.2 | $90.79 \pm 0.15$ |
| LSTM-2048-512 | — | 47.7 | N/A | $91.40 \pm 0.18$ |
| LSTM-2048-512 | LSTM-2048-512 | 47.7 | 47.3 | $91.62 \pm 0.23$ |
| CNN-BIG-LSTM | — | 30.0 | N/A | $91.66 \pm 0.13$ |
| CNN-BIG-LSTM | LSTM-2048-512 | 30.0 | 47.3 | $\mathbf{91.93 \pm 0.19}$ |

Table 6: Comparison of CoNLL-2003 test set $F_1$ for different language model combinations. All language models were trained and evaluated on the 1B Word Benchmark, except LSTM-512-256* which was trained and evaluated on the standard splits of the NER CoNLL 2003 dataset.

tion functions (i.e., the RNN parameters in the language model) from much larger data compared to the composition functions in the baseline tagger, which are only learned from labeled data.

**Importance of task specific RNN.** To understand the importance of including a task specific sequence RNN we ran an experiment that removed the task specific sequence RNN and used only the LM embeddings with a dense layer and CRF to predict output tags. In this setup, performance was very low, 88.17 $F_1$, well below our baseline. This result confirms that the RNNs in the baseline tagger encode essential information which are not encoded in the LM embeddings. This is unsurprising since the RNNs in the baseline tagger are trained on labeled examples, unlike the RNN in the language model which is only trained on unlabeled examples.

**Dataset size.** *A priori*, we expect the addition of LM embeddings to be most beneficial in cases where the task specific annotated datasets are small. To test this hypothesis, we replicated the setup from Yang et al. (2017) that samples 1% of the CoNLL 2003 training set and compared the performance of TagLM to our baseline without LM. In this scenario, test $F_1$ increased 3.35% (from 67.66 to 71.01%) compared to an increase of 1.06% $F_1$ for a similar comparison with the full training dataset. The analogous increases in Yang et al. (2017) are 3.97% for cross-lingual transfer from CoNLL 2002 Spanish NER and 6.28% $F_1$ for transfer from PTB POS tags. However, they found only a 0.06% $F_1$ increase when using the full training data and transferring from both CoNLL 2000 chunks and PTB POS tags. Taken together, this suggests that for very small labeled training sets, transferring from other tasks yields

a large improvement, but this improvement almost disappears when the training data is large. On the other hand, our approach is less dependent on the training set size and significantly improves performance even with larger training sets.

**Number of parameters.** Our TagLM formulation increases the number of parameters in the second RNN layer $R_2$ due to the increase in the input dimension $\mathbf{h}_1$ if all other hyperparameters are held constant. To confirm that this did not have a material impact on the results, we ran two additional experiments. In the first, we trained a system without a LM but increased the second RNN layer hidden dimension so that number of parameters was the same as in TagLM. In this case, performance *decreased* slightly (by 0.15% $F_1$) compared to the baseline model, indicating that solely increasing parameters does not improve performance. In the second experiment, we decreased the hidden dimension of the second RNN layer in TagLM to give it the same number of parameters as the baseline no LM model. In this case, test $F_1$ *increased* slightly to $92.00 \pm 0.11$ indicating that the additional parameters in TagLM are slightly hurting performance.[4]

**Does the LM transfer across domains?** One artifact of our evaluation framework is that both the labeled data in the chunking and NER tasks and the unlabeled text in the 1 Billion Word Benchmark used to train the bidirectional LMs are derived from news articles. To test the sensitivity to the LM training domain, we also applied TagLM with a LM trained on news articles to the SemEval 2017 Shared Task 10, ScienceIE.[5] Scien-

---

[4] A similar experiment for the Chunking task did not improve $F_1$ so this conclusion is task dependent.

[5] https://scienceie.github.io/

ceIE requires end-to-end joint entity and relationship extraction from scientific publications across three diverse fields (computer science, material sciences, and physics) and defines three broad entity types (Task, Material and Process). For this task, TagLM increased $F_1$ on the development set by 4.12% (from 49.93 to to 54.05%) for entity extraction over our baseline without LM embeddings.[6] We conclude that LM embeddings can improve the performance of a sequence tagger even when the data comes from a different domain.

## 4 Related work

**Unlabeled data.** TagLM was inspired by the widespread use of pre-trained word embeddings in supervised sequence tagging models. Besides pre-trained word embeddings, our method is most closely related to Li and McCallum (2005). Instead of using a LM, Li and McCallum (2005) uses a probabilistic generative model to infer context-sensitive latent variables for each token, which are then used as extra features in a supervised CRF tagger (Lafferty et al., 2001). Other semi-supervised learning methods for structured prediction problems include co-training (Blum and Mitchell, 1998; Pierce and Cardie, 2001), expectation maximization (Nigam et al., 2000; Mohit and Hwa, 2005), structural learning (Ando and Zhang, 2005) and maximum discriminant functions (Suzuki et al., 2007; Suzuki and Isozaki, 2008). It is easy to combine TagLM with any of the above methods by including LM embeddings as additional features in the discriminative components of the model (except for expectation maximization). A detailed discussion of semi-supervised learning methods in NLP can be found in (Søgaard, 2013).

LM embeddings are related to a class of methods (e.g., Le and Mikolov, 2014; Kiros et al., 2015; Hill et al., 2016) for learning sentence and document encoders from unlabeled data, which can be used for text classification and textual entailment among other tasks.

**Neural language models.** LMs have always been a critical component in statistical machine translation systems (Koehn, 2009). Recently, neural LMs (Bengio et al., 2003; Mikolov et al., 2010) have also been integrated in neural machine translation systems (e.g., Kalchbrenner and Blunsom,

---

[6]Test data set performance withheld to preserve anonymity and will be updated before publication.

2013; Devlin et al., 2014) to score candidate translations. In contrast, TagLM uses neural LMs to encode words in the input sequence.

Unlike forward LMs, bidirectional LMs have received little prior attention. Most similar to our formulation, Peris and Casacuberta (2015) used a bidirectional neural LM in a statistical machine translation system for instance selection. They tied the input token embeddings and softmax weights in the forward and backward directions, unlike our approach which uses two distinct models without any shared parameters. Frinken et al. (2012) also used a bidirectional n-gram LM for handwriting recognition.

**Interpreting RNN states.** Recently, there has been some interest in interpreting the activations of RNNs. Linzen et al. (2016) showed that single LSTM units can learn to predict singular-plural distinctions. Karpathy et al. (2015) visualized character level LSTM states and showed that individual cells capture long-range dependencies such as line lengths, quotes and brackets. Our work complements these studies by showing that LM states are useful for downstream tasks as a way of interpreting what they learn.

**Other sequence tagging models.** Current state of the art results in sequence tagging problems are based on bidirectional RNN models. However, many other sequence tagging models have been proposed in the literature for this class of problems (e.g., Lafferty et al., 2001; Collins, 2002). LM embeddings could also be used as additional features in other models, although it is not clear whether the model complexity would be sufficient to effectively make use of them.

## 5 Conclusion

In this paper, we proposed a simple and general semi-supervised method using pre-trained neural language models to augment token representations in sequence tagging models. Our method significantly outperforms current state of the art models in two popular datasets for NER and Chunking. Our analysis shows that adding a backward LM in addition to traditional forward LMs consistently improves performance. The proposed method is robust even when the LM is trained on unlabeled data from a different domain, or when the baseline model is trained on a large number of labeled examples.

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
