# Peer review of "Semi-supervised sequence tagging with bidirectional language models"

_ACL 2017 — decision unknown_

[Official Review · Reviewer 1 · rating 4 · confidence 4]
soundness 3 · originality 3 · clarity 4 · impact 4 · substance 4 · appropriateness 5 · meaningful comparison 4 · presentation format Oral Presentation

The paper introduces a general method for improving NLP tasks using embeddings
from language models. Context independent word representations have been very
useful, and this paper proposes a nice extension by using context-dependent
word representations obtained from the hidden states of neural language models.
They show significant improvements in tagging and chunking tasks from including
embeddings from large language models. There is also interesting analysis which
answers several natural questions.

Overall this is a very good paper, but I have several suggestions:
- Too many experiments are carried out on the test set. Please change Tables 5
and 6 to use development data
- It would be really nice to see results on some more tasks - NER tagging and
chunking don't have many interesting long range dependencies, and the language
model might really help in those cases. I'd love to see results on SRL or CCG
supertagging.
- The paper claims that using a task specific RNN is necessary because a CRF on
top of language model embeddings performs poorly. It wasn't clear to me if they
were backpropagating into the language model in this experiment - but if not,
it certainly seems like there is potential for that to make a task specific RNN
unnecessary.

[Official Review · Reviewer 2 · rating 3 · confidence 4]
soundness 3 · originality 3 · clarity 4 · impact 4 · substance 2 · appropriateness 5 · meaningful comparison 4 · presentation format Poster

The paper proposes an approach where pre-trained word embeddings and
pre-trained neural language model embeddings are leveraged (i.e., concatenated)
to improve the performance in English chunking and NER on the respective CoNLL
benchmarks, and on an out-of-domain English NER test set. The method records
state-of-the-art scores for the two tasks.

- Strengths:

For the most part, the paper is well-written and easy to follow. The method is
extensively documented. The discussion is broad and thorough.

- Weaknesses:

Sequence tagging does not equal chunking and NER. I am surprised not to see POS
tagging included in the experiment, while more sequence tagging tasks would be
welcome: grammatical error detection, supersense tagging, CCG supertagging,
etc. This way, the paper is on chunking and NER for English, not for sequence
tagging in general, as it lacks both the multilingual component and the breadth
of tasks.

While I welcomed the extensive description of the method, I do think that
figures 1 and 2 overlap and that only one would have sufficed.

Related to that, the method itself is rather straightforward and simple. While
this is by all means not a bad thing, it seems that this contribution could
have been better suited for a short paper. Since I do enjoy the more extensive
discussion section, I do not necessarily see it as a flaw, but the core of the
method itself does not strike me as particularly exciting. It's more of a
"focused contribution" (short paper description from the call) than
"substantial" work (long paper).

- General Discussion:

Bottomline, the paper concatenates two embeddings, and sees improvements in
English chunking and NER.

As such, does it warrant publication as an ACL long paper? I am ambivalent, so
I will let my score reflect that, even if I slightly lean towards a negative
answer. Why? Mainly because I would have preferred to see more breadth: a) more
sequence tagging tasks and b) more languages.

Also, we do not know how well this method scales to low(er)-resource scenarios.
What if the pre-trained embeddings are not available? What if they were not as
sizeable as they are? The experiments do include a notion of that, but still
far above the low-resource range. Could they not have been learned in a
multi-task learning setup in your model? That would have been more substantial
in my view.

For these reasons, I vote borderline, but with a low originality score. The
idea of introducing context via the embeddings is nice in itself, but this
particular instantiation of it leaves a lot to ask for.